# A Unet-based method for Cell Segmentation Challenge

**Lanfeng Zhong**
School of Mechanical and Electrical Engineering
University of Electronic Science and Technology of China
Chengdu, 611731, China
`lanfz@std.uestc.edu.cn`

**Meng Han**
School of Mechanical and Electrical Engineering
University of Electronic Science and Technology of China
Chengdu, 611731, China

## Abstract

Automatic detection and segmentation of cells and nuclei in microscopy images is important for many biological applications. The development of automated methods for nuclear segmentation and classification enables the quantitative analysis of tens of thousands of nuclei within a whole-slide microscopy image. In situations of crowded cells, some existing methods can be prone to segmentation errors, such as falsely merging bordering cells or suppressing valid cell instances due to the poor approximation with bounding boxes. For this challenge that contains more than one modality and are diverse in cell shape and color, we use a unet-based method to tackle this problem. The modified Unet can generate probability maps and distance maps in one forward step. We then utilize the probability maps and distance maps to obtain the final cell instance segmentation results.

## 1   Introduction

Many biological tasks rely on the accurate detection and segmentation of cells and nuclei from microscopy images. Examples include high-content screens of variations in cell phenotypes, or the identification of developmental lineages of dividing cells. In many cases, the goal is to obtain an instance segmentation, which is the assignment of a cell instance identity to every pixel of the image. To that end, a prevalent bottom-up approach is to first classify every pixel into semantic classes (such as cell or background) and then group pixels of the same class into individual instances. The first step is typically done with learned classifiers, such as random forests or neural networks. Pixel grouping can for example be done by finding connected components. While this approach often gives good results, it is problematic for images of very crowded cell nuclei, since only a few mis-classified pixels can cause bordering but distinct cell instances to be fused.

An baseline method is to first segment the cell, and then transform the binary masks into instance maps via specific post-processing. An alternative top-down approach is to first localize individual cell instances with a rough shape representation and then refine the shape in an additional step.

To alleviate the aforementioned problems, we use StarDist [1], a cell detection method that predicts a shape representation which is flexible enough such that - without refinement - the accuracy of the localization can compete with that of instance segmentation methods.

36th Conference on Neural Information Processing Systems (NeurIPS 2022).

## 2 Method

The pipeline of the method is shown as Fig. The approach is similar to object detection methods that directly predict shapes for each object of interest. Unlike most of them, we do not use axis-aligned bounding boxes as the shape representation. Instead, the model predicts a star-convex polygon for every pixel. The stardist model generates two outputs: probability map for cell region, and distance maps. At post-processing, a module named non-maximum suppression (NMS) is used to obtain the final instance results.

### 2.1 Preprocessing

We perform normalization for all images, including training and validation images.

### 2.2 Stardist

While we could simply classify each pixel as either object or background based on binary masks, we instead define its object probability $d_{i,j}$ as the (normalized) Euclidean distance to the nearest background pixel. Only use the object probability information as supervision is not enough to train a robust model for instance segmentation. For instance labels, there is some additional information we can obtain, such as centroid, outline and distance maps. We can utilize these different types of information as supervision to train a network. For every pixel belonging to an object, the Euclidean distances $r_{i,j}^k$ to the object boundary can be computed by simply following each radial direction k until a pixel with a different object identity is encountered.

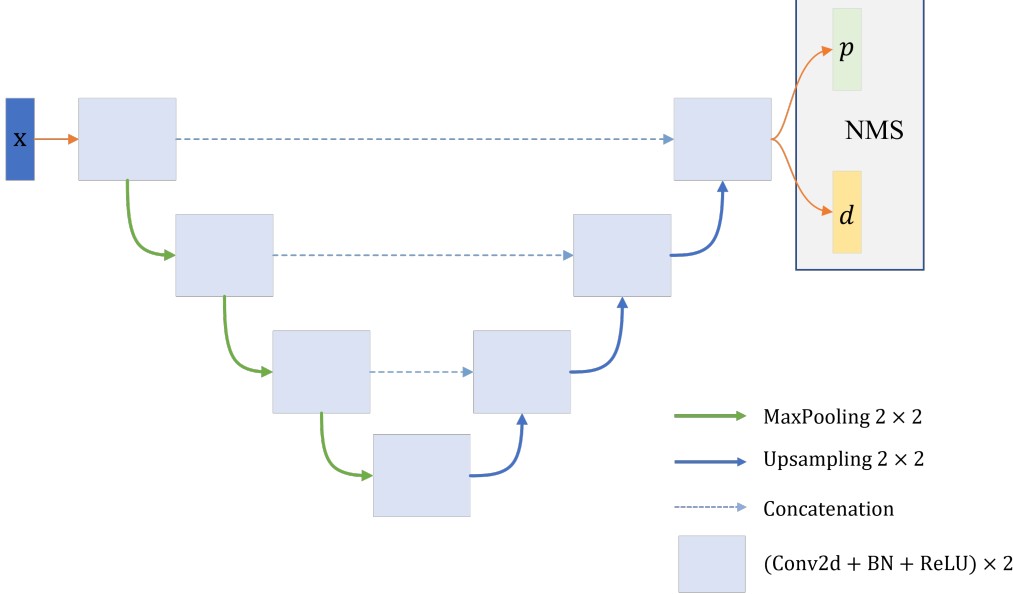

Figure 1: Unet-based Stardist architecture. The model generates two outputs, and utilize both of them to obtain the instance results.

We aims to use CPS [2] as our strategy to use the unlabelled cases, however, we have no time to implement the combination between the stardist and CPS.

**Loss function.** we use the summation between Dice loss and cross entropy loss because compound loss functions have been proved to be robust in various medical image segmentation tasks [3]. For distance maps and corresponding ground truth, we use MAE loss.

### 2.3 Post-processing

**Non-maximum suppression.** We perform common, greedy non-maximum suppression (NMS) to only retain those polygons in a certain region with the highest object probabilities. We only consider

polygons associated with pixels above an object probability threshold as candidates, and compute their intersections with a standard polygon clipping method.

# 3   Experiments

## 3.1   Implementation details

Please include all the implementation details in this subsection. The following items are minimal requirements.

### 3.1.1   Environment settings

The development environments and requirements are presented in Table 1.

Table 1: Development environments and requirements.

| System | Ubuntu 20.04.5 LTS |
|---|---|
| CPU | i7-6850K CPU @ 3.60GHz 3.60 GHz |
| RAM | 64GB; 2.67MT/s |
| GPU | One NVIDIA 2080Ti 11G |
| CUDA version | 11.4 |
| Programming language | Python 3.8 |
| Deep learning framework | Tensorflow-gpu 2.7.0 |
| Specific dependencies | None |
| Code | Will be available soon |

### 3.1.2   Training protocols

**Data augmentation**

Patch sampling strategy during training (e.g., random sample $512 \times 512$ patches) and inference (slide window with a patch size $512 \times 512$). We also use random flip, random intensity change (with intensity parameter of 0.3) and random Gaussian noise. In these data augmentations, the random intensity change has the best performance.

We randomly split the 1000 labelled images into 900, 100 for training and validation, respectively. We select the best model in the validation set and use it to generate tuning set results.

Table 2: Training protocols. If the method includes more than one model, please present this table for each model seperately.

| Network initialization | "kaiming" normal initialization |
|---|---|
| Batch size | 10 |
| Patch size | 512×512 |
| Total epochs | 600 |
| Optimizer | SGD with nesterov momentum ($\mu = 0.99$) |
| Initial learning rate (lr) | 0.001 |
| ReduceLROnPlateau,patience: 30, factor: 0.5 | |
| Training time | 10 hours |
| Loss function | Dice and CE loss, MAE loss |
| Number of model parameters | 2.89M[1] |
| Number of flops | 48.84G[2] |

# 4   Results and discussion

In this challenge, we did not exploit the unlabelled data.

The method performs well in the cell images that have regular shape, while badly in the irregular cell images.

The possible reason for the failed cell images is that the hypothesis of stardist localizes cell nuclei via star-convex polygons. It will not localize well in those cell images with irregular shape.

For whole-slide image, we use slide-window reference. As the model only produce the instance label for each patch, we rearrange the instance number for the whole-slide image after sticking the patch into the original image.

## 4.1 Quantitative results on tuning set

In tuning set, there are 101 images in total. We get the f1 score of 0.6551 as our result.

We did not use the unlabelled data.

## 4.2 Qualitative results on validation set

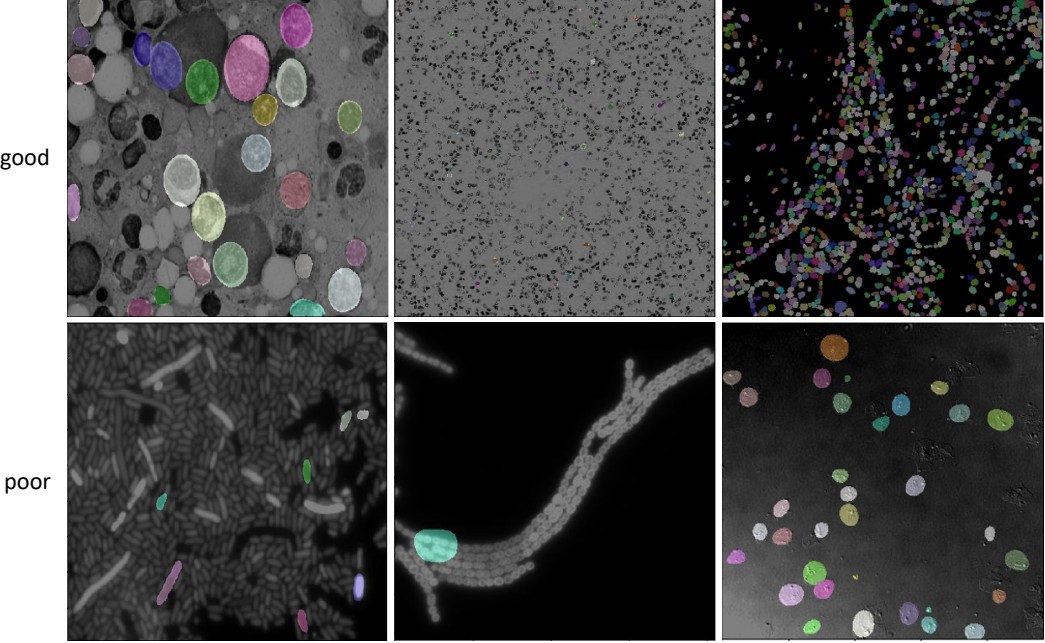

Figure 2: The first row shows three good examples. The second row shows three poor examples.

Qualitative results on validation set is shown in Fig. 2. Fig. 2 shows three good examples and three poor examples, respectively. It can be seen that stardist can distinguish very crowded cells. However, the stardist model does not have the generalization ability for irregular cells, leading to poor performance of some cases.

## 4.3 Results on final testing set

This is a placeholder. We will send you testing results after the challenge.

## 4.4 Limitation and future work

The stardist model functions well in the cell images with regular shape, while it can performs badly in the cell images with irregular shape.

# 5    Conclusion

We demonstrated that star-convex polygons are a good shape representation to accurately localize cell nuclei even under challenging conditions. The stardist model is especially appealing for images of very crowded cells. However, it has one main defect: can not segment irregular cells.

## Acknowledgement

The authors of this paper declare that the segmentation method they implemented for participation in the NeurIPS 2022 Cell Segmentation challenge has not used any private datasets other than those provided by the organizers and the official external datasets and pretrained models. The proposed solution is fully automatic without any manual intervention.

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
