# OpenReview forum: "A Unet-based method for Cell Segmentation Challenge"
_NeurIPS.cc/2022/Challenge/CellSeg — Submitted to NeurIPS CellSeg 2022_

### Official Review · Program_Chairs · 2023-01-15
**the performance is good but the writing should be significantly improved**

**Rating:** 5
**Confidence:** 5

**Review:**


The paper is poor writing and too short. The length should be extended to at least 8 pages.

1. Many details are missed in the method and results.

2. "We have no time..." It has been two months after the testing submission deadline. Thus, time shouldn't be a reason for detailed analysis in the revision (e.g., using unlabeled data, external data; ablation study...)

---

### Decision · Program_Chairs · 2023-01-19

Reject